# Distribution Characteristics of the Geoelectric Field in Waste Dump Slopes during the Evolution of Instability Sources under Rainfall Conditions

**Ensheng Kang** [1,*] , **Haidong Meng** [1,*], **Zexi Zhao** [2] and **Zihao Zhao** [1]

1 School of Mining and Coal, Inner Mongolia University of Science and Technology, Baotou 014010, China
2 College of Resources and Civil Engineering, Northeastern University, Shengyang 110057, China
* Correspondence: kangensheng@imust.edu.cn (E.K.); haidongm@imust.edu.cn (H.M.)

**Abstract:** To study the evolution of geological hazard sources of waste dump slopes under rainfall conditions, a physical model of a rainfall-affected slope was designed. The apparent resistivity of the slope rock and soil mass at different rainfall times was measured via the high-density resistivity method, and the formation process of internal disaster sources of the rainfall-affected slope was obtained. The variation characteristics of the resistivity of the rain-affected slope were analyzed when it had a weak surface and crack development. Based on the three-water model and Maxwell conductivity formula, the evolution process of geological hazard sources of the rainfall-affected slope was summarized. A resistivity response mechanism equation for rainfall-induced slope hazard sources was derived and compared to the Archie formula, verifying the model rationality. The test results showed that the behavior of the rainfall-affected slope conforms to the saturated–unsaturated dynamic cycle process. The apparent resistivity was positively correlated with the development of slope pores and cracks and negatively correlated with the water content in the slope. The apparent resistivity increased during fracture development and decreased during water seepage. In the slope failure and disaster process, the apparent resistivity varies under the coupling effect of crack development and water seepage. During the formation of geological hazard sources, the apparent resistivity abruptly changes and fluctuates. Therefore, according to the abrupt changes and abnormal fluctuations in the apparent resistivity detected, the development of geological hazard sources of slopes can be determined.

**Keywords:** waste dump; slope; rainfall; seepage; geological disaster sources; resistivity



## 1. Introduction

The rainy season exhibits a high incidence of geological slope disasters. Geological slope disasters such as debris flows and landslides are readily formed, especially under continuous rainfall [1–4]. Research has shown that in the rainfall seepage process, the rock and soil mass bulk density and the pore water pressure increase, the sliding force of the slope body increases, and the shear strength decreases, resulting in slope failure [5,6]. Cracks and weak planes formed during rainfall and seepage are the main geological disaster sources driving slope failure [7,8]. Determination of the geological conditions and change trends of the slope structure, lithology, crack development, water content and weak planes is essential for effective slope disaster prediction and forecasting.

Scholars have widely researched the geological hazards of rainfall-affected slopes. Liu et al. [9] used geostatistical techniques to study the geological strength index distribution in mine slope rock masses and analyzed the correlation between the spatiotemporal variability characteristics of rock mechanical parameters and the development of geological slope hazards. Kang et al. [10] employed comprehensive geophysical exploration techniques, namely the high-density electrical method and geological radar method, to survey the factors influencing the instability of open pit mine slopes. The impact of groundwater

seepage on the stability of mine slopes was analyzed. Liang et al. [11] found that the generation of cracks in the rainfall seepage process is the key factor inducing slope instability, and most landslides occur during or shortly after rainfall. Zhou et al. [12] studied the progressive failure mode and water transport behavior of slopes under rainfall seepage and examined the slope disaster mechanism controlled by cracks. The research results showed that the slope stability decreases with increasing initial crack depth and gradually decreases with increasing soil saturation ratio, finally resulting in a slope disaster. Wang et al. [13] investigated the response relationship between the water content in slope soil, pore water pressure, displacement and the rainfall process, infiltration trend and disaster mode through physical model tests. The change in the soil moisture content in the rainfall-affected slope exhibited a lag, and the slope displacement was significantly positively correlated with the change rate of the water content. The mechanism of slope disasters can be divided into three processes: scattered collapse of the steep slope, formation of tension cracks in the slope and large-scale collapse.

Due to its advantages of rapid and nondestructive detection, the high-density resistivity method has been widely adopted in the study of underground media. Ismail et al. [14] studied the underground resistivity distribution in a slope and mapped the potential risk areas of geological slope hazards. Weak zones were characterized by a large proportion of high water-saturation zones and low resistivity values. Liu et al. [15] established a soil resistivity test method for conventional and supergravity environments based on the high-density electrical method, studied the correlation between the seepage soil resistivity and water content, and verified the effectiveness of the high-density electrical method in the detection of soil moisture migration. Yan et al. [16] examined the relationship between the water content and resistivity of slope soil by combining the high-density electrical method and time-domain reflection technique, revealed the distribution characteristics and migration trend of slope groundwater, and realized rapid and nondestructive detection of slope water migration. Ou et al. [17] studied the trend of soil crack development via high-density resistivity imaging technology, verifying that the crack development time is earlier than that of mining face formation. Li et al. [18] performed electrical measurement tests, triaxial compression tests and small-scale tests of slope soft layer materials under different water content conditions, studied the resistivity–water content relationship and resistivity change rate-stress-strain relationship, and preliminarily established the relationship between slope soft layer damage and resistivity change. Relevant research conclusions have validated the feasibility of describing the internal structural changes in rock and soil masses with geoelectric field parameters. However, research on the evolution trend of geological disaster sources of waste dump slopes under the influence of rainfall is lacking.

In this paper, a test platform for waste dump slopes under rainfall conditions was designed and manufactured. The high-density resistivity method was adopted to detect the change in the slope resistivity in the rainfall seepage process. The geoelectric field response characteristics of slope weak planes and crack evolution during seepage flow were studied. We explored the multisource information of geological slope disaster source evolution and disaster warning, which could provide support for the analysis of rainfall-related waste dump slope disaster mechanisms and disaster prediction. In this article, the research results are examined from several aspects: (1) experimental materials and methods, (2) characteristics and trends of electrical resistivity changes in waste dumps during rainfall, (3) electrical resistivity responses and evolution mechanisms of disaster sources caused by rainfall, and (4) distribution characteristics of the geoelectric field formed by disaster sources.

## 2. Materials and Methods

### 2.1. Overview of the Prototype Slope

Both the prototype slope bench height and safety platform width are 20 m. The main components of the slope include silty clay, clay, medium-coarse sand, gravel, slate, and mica amphibolite. Table 1 provides the physical and mechanical parameters.

**Table 1.** Physical and mechanical parameters of the prototype slope.

| Sedimentation Coefficient (%) | Slope Angle (°) | Cohesive Forces (kPa) | Angle of Internal Friction (°) |
|---|---|---|---|
| 15 | 32 | 36.5 | 38.4 |

### 2.2. Test Model System

The test focuses on the development of weak planes and cracks in the slope survey line area. By photographing cracks at the top of the slope, the water content in the rock and soil mass and the resistivity can be measured in the water migration and crack development process, and the evolution trend of geological disaster sources of rainfall-affected slopes can be studied.

We designed and constructed a physical model of a waste dump slope under rainfall conditions. The model system includes the test slope, rainfall equipment, photographic equipment, water content measurement equipment and resistivity measurement equipment. Table 2 shows the parameters of the slope model.

**Table 2.** Parameters of the slope model.

| Slope Parameters | Value |
|---|---|
| Length (m) | 3 |
| Width (m) | 2 |
| Height (m) | 0.7 |
| Width of slope top (m) | 0.75 |
| Slope angle (°) | 32 |
| Rainfall intensity (mm/min) | 0.21 |

The resistivity measurement equipment is a DUK-2A high-density electrical instrument manufactured by CGE (Chongqing city, China) Geological Instrument Co., Ltd. The custom-made copper electrode is 3.5 cm long and 7 cm wide. A total of 30 electrodes were set. The electrode arrangement and measurement method of the Wenner device were adopted. The maximum isolation coefficient is 9, and the minimum isolation coefficient is 1. The resistivity change diagram could be obtained by inversion using Res2dinv software. The test model system is shown in Figure 1.

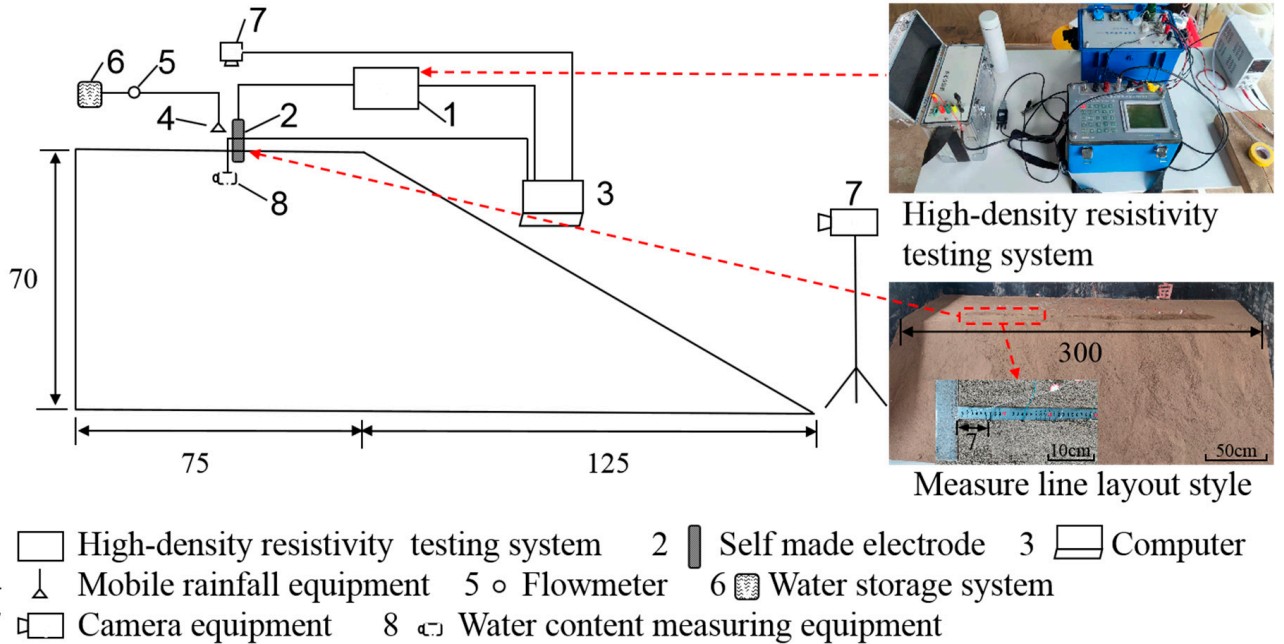

**Figure 1.** Design diagram of test model (unit: cm).

The test materials were mixed according to the parameters provided in Tables 3 and 4. The test slope was constructed in layers, and each layer was compacted before the next layer was emplaced. Table 3 shows the dry density, wet density, mixed density, and mixing ratio of the different materials.

**Table 3.** Parameters of test materials.

| Type | Dry Density (g·cm$^{-3}$) | Wet Density (g·cm$^{-3}$) | Permeability Coefficient (cm·s) | Mass Ratio of Materials |
|---|---|---|---|---|
| Soil | 1.231 | 1.657 | | 5 |
| Sand | 1.339 | 1.725 | | 3 |
| Small grain stone | 1.347 | 1.949 | | 1 |
| Mixture | 1.295 | 1.836 | $4.06 \times 10^{-5}$ | |

**Table 4.** Mass fraction of each particle size of test materials.

| Number | Particle Size (d/mm) | Mass Fraction (%) |
|---|---|---|
| 1 | >2 | 3.1 |
| 2 | 2—1 | 8.7 |
| 3 | 1—0.5 | 27.8 |
| 4 | 0.5—0.125 | 26.7 |
| 5 | 0.125—0.075 | 32.5 |
| 6 | <0.075 | 4.3 |

Table 4 provides the mass fraction of the test materials for each particle size. During the formation of the prototype waste dump slope, the distribution pattern and original mechanical structure of the rock and soil mass are destroyed. The particle size of the slope increases from top to bottom. The internal structure of the slope is continuous before rainfall, without faults, cracks and weak planes. The porosity of the rock and soil mass is higher than that of the original rock, and the apparent resistivity is higher. Affected by

compaction and dead weight, the apparent resistivity in shallow areas is higher than that in deep areas, and high-resistivity abnormal areas occur in some parts.

*2.3. Study Protocol*

Before model construction, the test materials were screened and classified, and the dry and wet densities and permeability coefficient of the classified materials (by the particle size) and mixed rock-soil samples were measured. Before the test, many preliminary tests were conducted. The shoulder and front edge of the top of the waste dump slope are greatly affected by the free surface under the influence of rainfall seepage, which is the main occurrence area of cracks. Therefore, a resistivity measuring line was arranged along the horizontal direction at the top of the slope.

The test was conducted during the cycle of concentrated rainfall and measurement after rainfall cessation. In the rainfall process, the rainfall intensity and location of the rainfall devices should be controlled to ensure that rainwater can fully penetrate into the slope without large-scale runoff. The apparent resistivity beneath the survey line was measured using high-density resistivity equipment after rainfall cessation. The rainfall and measurement process was repeated until the slope was saturated. The test was stopped when a geological slope hazard occurred. By observing the inversion of the change in the resistivity inside the slope model and comparing it to the development of cracks, the resistivity characteristics and change trends could be analyzed in the process of rainfall seepage and crack development, and the development process of geological disaster sources of the rainfall-affected slope was studied.

## 3. Fundamentals

### 3.1. Saturated–Unsaturated Seepage Flow Theory

Infiltration in rainfall-affected slopes is a saturated–unsaturated seepage process, and the seepage differential formula can be expressed as [19,20]:

$$\frac{\partial}{\partial x}\left[k_{wx}\frac{\partial H}{\partial x}\right] + \frac{\partial}{\partial y}\left[k_{wy}\frac{\partial H}{\partial y}\right] + Q = \frac{\partial \theta}{\partial t} = m_w \gamma_w \frac{\partial H}{\partial t} \tag{1}$$

where $H$ is the total head, m; $k_{wx}$ and $k_{wy}$ are the permeability coefficients along the $x$ and $y$ directions, respectively, m/s; $Q$ is the water infiltration amount in the soil mass, L/s; $\theta$ is the volumetric moisture content in the soil mass; $t$ is the seepage time, s; $\gamma_w$ is the unit weight of water; and $m_w$ is the slope of the soil water characteristic curve.

### 3.2. Direct Current (DC) Resistivity Theory

The DC resistivity method describes the underground structure through the difference between the rock and soil conductivities. In engineering practice, the apparent resistivity ($\rho_S$) is often used to describe the comprehensive state of underground uneven materials and the topographic relief. The power supply electrodes A and B transmit DC electrical signals into the ground, the potential difference between any two electrodes M and N is measured and the apparent resistivity of the medium can be obtained [15]. The most frequently used DC resistivity methods include electrical sounding, electrical profiling and high-density resistivity methods. The measurement principle of the DC resistivity method is shown in Figure 2.

$$\rho_S = K\frac{\Delta U_{MN}}{I} \tag{2}$$

where $K$ is the device coefficient, $\Delta U_{MN}$ represents the potential difference between the measuring electrodes $M$ and $N$, and $I$ represents the power supply current.

$$K = \pi \frac{AM \cdot AN}{MN} \tag{3}$$

$$\Delta U_{MN} = \int_N^M E_{MN} \cdot dl = \int_N^M j_{MN} \cdot \rho_{MN} dl \tag{4}$$

where $E_{MN}$ is the electric field strength between the measuring electrodes $M$ and $N$, $j_{MN}$ is the current density between the measuring electrodes $M$ and $N$, $\rho_{MN}$ is the resistivity between the measuring electrodes $M$ and $N$, and $dl$ is the length unit between the measuring electrodes $M$ and $N$.

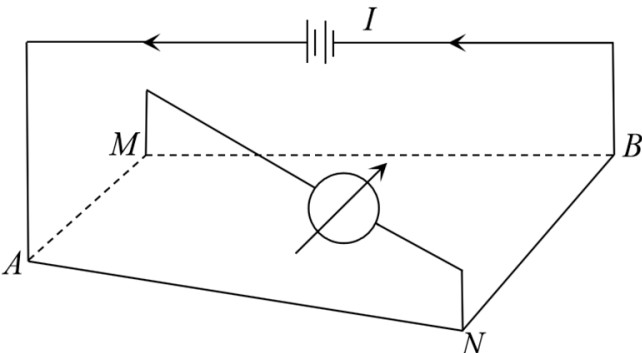

**Figure 2.** Measuring principle of DC resistivity method.

The Wenner device was symmetrically installed with four poles. The electrode arrangement and measurement method are shown in Figure 3. Considering $AM = MN = NB = $ a, Equation (3) can be expressed as:

$$K = 2\pi a \tag{5}$$

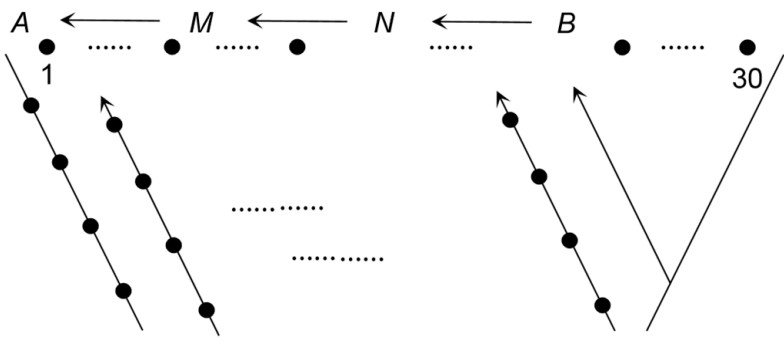

**Figure 3.** Wenner configuration and cross-section scanning diagram.

By simultaneously solving Equations (2), (4) and (5), the resistivity can be expressed as

$$\rho_S = \frac{K}{I} \int_N^M E_{MN} \cdot dl = \frac{2\pi a}{I} \int_N^M j_{MN} \cdot \rho_{MN} dl \tag{6}$$

Research has shown that there is a negative correlation between the resistivity and water content in rock and soil masses, which can be used to characterize water migration in rock and soil masses through the change in resistivity [21,22]. The resistivity of the rock and soil mass is positively correlated with the porosity, and the resistivity can thus be employed to characterize the development process of rock and soil cracks [23–25]. During rainfall, the occurrence form of water in the slope rock and soil mass changes with the change in the water content, and the apparent resistivity constantly varies. Therefore, it is feasible to describe the change within the slope interior during rainfall through the change in the apparent resistivity.

### 3.3. Resistivity Inversion Theory

Assume that the apparent resistivity inversion model comprises several rectangular blocks with a constant resistivity, and the apparent resistivity of each rectangular block can be determined via the iterative nonlinear optimization method. In this paper, the apparent resistivity value was inverted via the least square method under the smooth restricted condition, as expressed in Equation (7) [26].

$$\left(J^T J + \lambda W^T W\right) \widetilde{P} = J^T \widetilde{g} \tag{7}$$

where $J$ is the Jacobian partial derivative matrix, $\lambda$ is the damping coefficient, $W$ is the smoothing filter parameter matrix and $\widetilde{P}$ is the correction vector of model parameters, $\widetilde{g}$ is the deviation vector of the logarithmic difference between the measured apparent resistivity and the calculated apparent resistivity.

The time-delay resistivity inversion function describes the change in the geoelectric parameters of the underground medium at the same location at different times through time constraints, which can highlight the deformation and failure characteristics of the underground medium. The inversion results of the initial resistivity data were used as a reference for the subsequent inversion, and regularization in the space and time domains was introduced to improve the inversion speed and imaging accuracy [27]. The inversion objective function can be expressed as Equation (8), and the solution of the set of linear equations of the least squares method can be obtained as:

$$\left(J_i^T R_d J_i + \lambda_i W^T R_m W\right) \Delta r_i^k = J_i^T R_d g_i - \lambda_i W^T R_m W r_{i-1}^k - \beta_i \lambda_i W^T R_i V \left(r_{i-1}^k - r_{i-1}^0\right) \tag{8}$$

where $J_i$ is the Jacobian partial derivative matrix, $i$ is the number of iterations, $g_i$ is the logarithmic difference matrix between the measured apparent resistivity and the calculated apparent resistivity, $W$ is the smoothing filter parameter matrix, $\lambda$ is the damping coefficient, $R_d$ and $R_m$ are the weight matrices, $r_{i-1}^0$ and $r_{i-1}^k$ is the initial value of the model and the k-th time value, $\Delta r_i^k$ is the model residual of the ith iteration, V is the weight matrix between models at different times and $\beta$ is the weight coefficient [28].

Zhang [29] introduced the residual value $\Delta\rho$ and resistivity reflection coefficient R to explain the change in the underground medium, as expressed in Equations (9) and (10), respectively. When $\Delta\rho$ is positive, the higher the value is, the higher the degree of crack development. When $\Delta\rho$ is negative, the higher the value is, the higher the water content in the medium. For $\rho_0 \to \infty$ and $R \to -1$, the current tends to occur within the medium $\rho_i$ range, while an increase in the $|R|$ value indicates an increase in the water content in the medium. For $\rho_0 \to 0$ and $R \to 1$, the current tends to occur within the medium $\rho_0$ range, while an increase in the $|R|$ value indicates that the water content in the medium decreases or cracks develop.

$$\Delta\rho = \rho_i - \rho_0 \tag{9}$$

$$R = \frac{\rho_i - \rho_0}{\rho_i + \rho_0} 100\% \tag{10}$$

where $\rho_i$ and $\rho_0$ are the resistivity value at time i and reference time, respectively.

## 4. Results and Analysis

### 4.1. Geoelectric Field Characteristics under Disaster Source Evolution in the Survey Line Area

The structure of the slope rock and soil mass changes during the wetting and drying cycle and is gradually destroyed to generate geological disaster sources such as cracks or aquifers. In the drying process, the porosity of the rock and soil mass decreases, and cracks are produced. In the wetting process, the porosity increases, and cracks decrease. When the water discharge rate is lower than the infiltration rate, an aquifer is formed within the slope [24].

Typical sections were selected to describe the characteristics of disaster source development. The setting of the monitoring sections is shown in Figure 4. Sections A–G correspond to depths of 0.875, 1.05, 1.085, 1.12, 1.155, 1.19 and 1.365 m. The resistivity–rainfall sequence relationship curve of the rainfall-affected waste dump slope is shown in Figures 5 and 6.

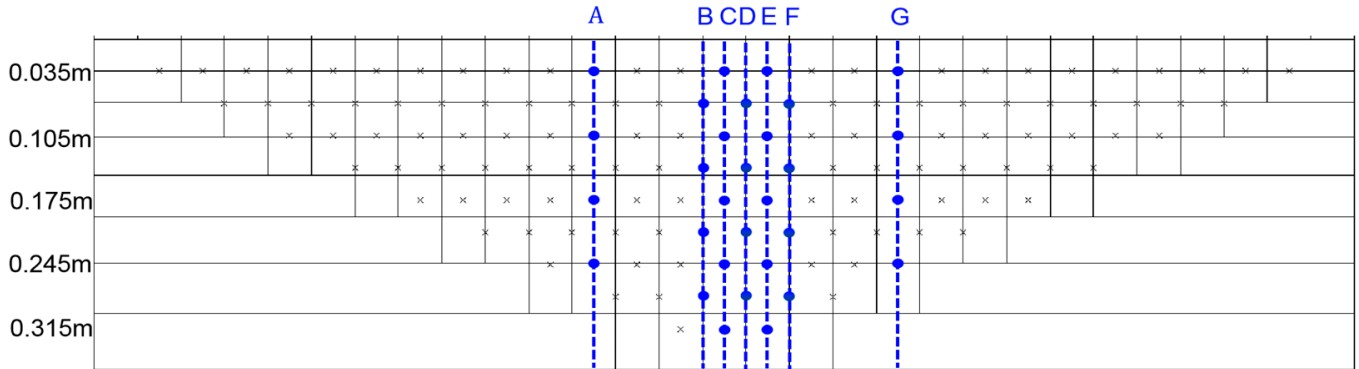

**Figure 4.** Location of monitoring section.

**(a)** 0.035 m depth

**(b)** 0.105 m depth

**(c)** 0.175 m depth

**(d)** 0.245 m depth

**Figure 5.** The variation characteristics of resistivity with rainfall period in the slope at sections A, E and G.

From the overall change trend, the apparent resistivity of the slope at each depth showed a downward trend in the rainfall process, namely, the apparent resistivity of the slope decreased with increasing rainfall process and water infiltration and finally tended to stabilize. With increasing depth, the time of apparent resistivity decline correspondingly increased. In the process of apparent resistivity decline, obvious abrupt changes and fluctuations occurred. Relevant research has shown that the influencing factors of rock and soil resistivity include the porosity, water content, saturation, temperature, water composition, rock and soil composition and structure, among which the porosity and saturation are the main influencing factors [25,30]. Under the test conditions, the temperature, water composition and geotechnical composition were fixed parameters, and the slope resistivity varied under the influences of the porosity, water content, saturation and internal structure.

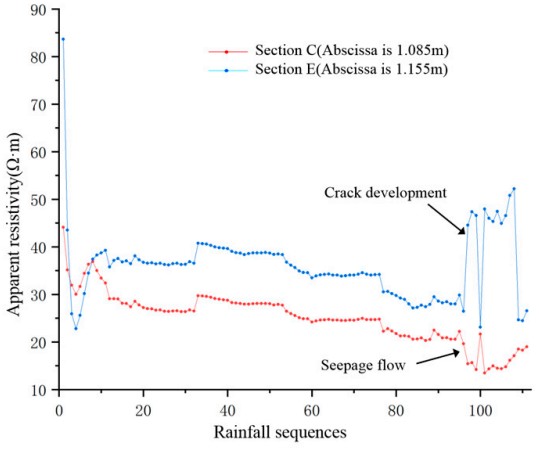

(**a**) 0.035 m depth

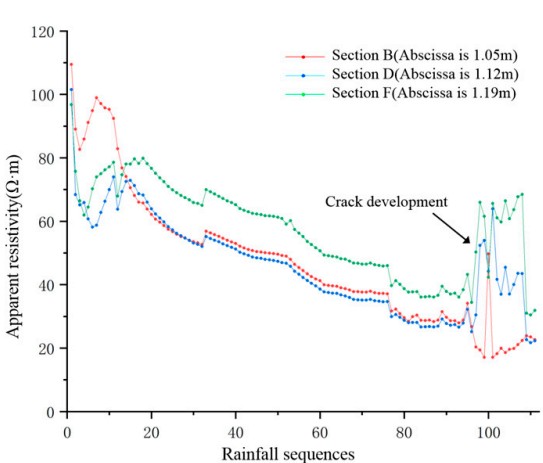

(**b**) 0.07 m depth

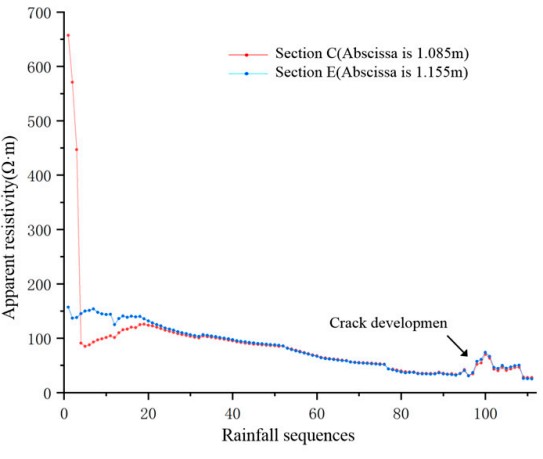

(**c**) 0.105 m depth

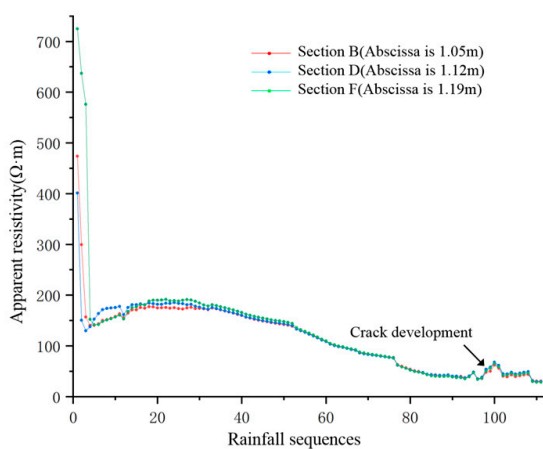

(**d**) 0.14 m depth

**Figure 6.** *Cont*.

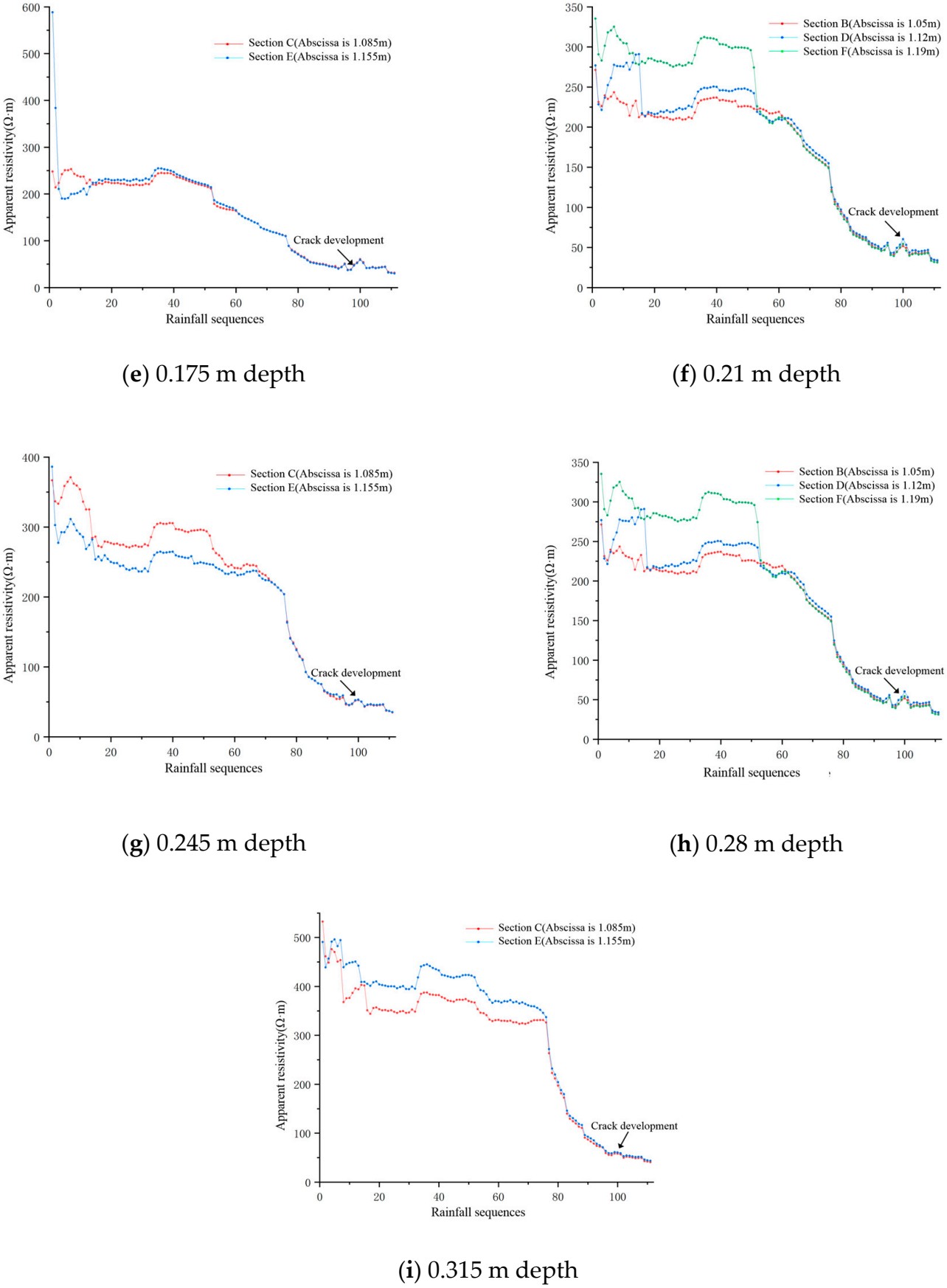

(**e**) 0.175 m depth

(**f**) 0.21 m depth

(**g**) 0.245 m depth

(**h**) 0.28 m depth

(**i**) 0.315 m depth

**Figure 6.** Resistivity-rainfall sequence diagram of sections B, C, D, E and F.

Figure 5 shows the relationship between the resistivity and rainfall sequence of sections A, E and G. As shown in Figure 5, after the fifth rainfall process, the apparent resistivity of each layer of the slope exhibited a significant sudden decline for the first time. The apparent resistivity of each layer slightly increased after decreasing. The apparent resistivity in the area at a depth of more than 0.035 m decreased to approximately 40 Ω·m, and ponding occurred at the top of the slope, indicating that the shallow slope rock and soil mass was nearly saturated. The sudden decline in the apparent resistivity at the beginning of rainfall occurred because with rainwater infiltration, the pores of the shallow slope rock and soil mass were rapidly filled with water, and conductive ions in the rock and soil entered the pore water. At the same time, because the conductivity of water is far higher than that of the rock and soil, the conductivity in the shallow area rapidly increased, and the apparent resistivity rapidly decreased. Affected by the increase in the saturated gravity of the upper slope, the deep rock and soil mass was compacted, the connectivity and conductivity were enhanced and the apparent resistivity in the deep area was reduced. At the measurement stage (after rainfall cessation), the loss of water from the slope resulted in uneven shrinkage of the rock and soil mass pores. When the shrinkage stress exceeded the tensile strength of the rock and soil mass, shrinkage cracks could form. In the rainfall process, water absorption and expansion of the rock and soil mass resulted in a loose arrangement of particles and an increase in the pore volume. When the expansion stress exceeded the tensile strength of the rock and soil mass, expansion cracks could occur [31–33]. The development of cracks led to a decrease in the electrical conductivity of the rock and soil mass and an increase in the potential difference, manifested as the increase in the resistance in the crack area and the apparent resistivity increased. That is, the resistivity in the fracture area increased with crack development [34,35].

After the 96th rainfall event, the apparent resistivity of section E of the shallow slope from 0.035 m to the top of the slope sharply increased and then fluctuated. The maximum apparent resistivity is 45 Ω·m, and the range (R) is 18 Ω·m. The apparent resistivity of section G from 0.105 m to the top of the slope suddenly increased and fluctuated after the 100th rainfall event. The maximum apparent resistivity is 144 Ω·m, and the range (R) is 98 Ω·m. According to the observation results, obvious cracks could be observed in the area of electrodes No. 16–18 (near section E) at the top of the slope after the 100th rainfall process. The Archie formula indicates that the resistivity (ρ) of water-bearing rock can be described by the porosity and saturation, as expressed in Equation (11) [36].

$$\rho = a\rho_w \varphi^{-\alpha} S^{-\beta} \tag{11}$$

where $\alpha$, $\beta$ and a are constants, $\varphi$ is porosity and $S$ is the volume fraction of pores filled with water, and $\rho_w$ is the resistivity of the pore fluid.

Under the same seepage medium conditions, the apparent resistivity was negatively related to the porosity and saturation within the slope. Therefore, the fluctuation in the apparent resistivity of section E after the 96th rainfall event shows that the porosity and saturation within the slope constantly varied, the internal structure of the slope in section E area was damaged, and cracks were formed.

According to the analysis of Figure 5 and fissure development in the slope top, the apparent resistivity data of each layer of sections B, C, D, E and F were selected, the apparent resistivity rainfall sequence diagram was generated, and the change characteristics of the apparent resistivity in the crack development process were analyzed, as shown in Figure 6.

Figure 6a,b show that after the 96th rainfall process, the apparent resistivity of sections E and F first increased and then fluctuated, and the maximum range (R) at the 0.035 m depth in section E reached 40 Ω·m. This indicates that cracks were first generated in the shallow slope near section E. Figure 6c–e reveals that after the 96th rainfall event, the resistivity values of the other layers also exhibited sudden changes of different amplitudes, and the change amplitude decreased with increasing depth. Obvious cracks appeared at the measuring line at the top of the slope after the 100th rainfall event. The results showed that corresponding areas within the slope had been damaged to varying degrees before the

formation of cracks in the slope top. In other words, after the 96th rainfall process, cracks were formed in the shallow slope and gradually developed towards the depths and surface of the slope. During the measurement (after rainfall cessation), under the action of water evaporation and seepage, the water content in the rock and soil mass above the critical depth continuously decreased, and the soil shrank and was deformed. When the tensile stress generated by shrinkage exceeded the tensile strength of the rock and soil mass and the lateral stress of the dead weight, the slope produced large shrinkage cracks, and the apparent resistivity dramatically increased. When rainfall again occurred, the rock and soil mass absorbed water and expanded. When the expansion stress exceeded the tensile strength and the lateral stress of the self-weight, shallow and short expansion cracks were generated, and the apparent resistivity value slightly increased. In the process of the drying and wetting cycle, crack development and water infiltration alternated along the cracks, reflected as the fluctuation in the apparent resistivity in the corresponding area. As the deep slope was less notably affected by water at the beginning of rainfall, the change form of the apparent resistivity differed from that in the shallow area, showing a slow fluctuation and decline. Figure 6f–i shows that after the 60th rainfall event, the apparent resistivity of the slope below 0.21 m sharply decreased and gradually approached the apparent resistivity value in the saturated state. After the test, the slope was excavated along the range from sections B to F. It was verified that the slope soil mass was gradually saturated in the rainfall and seepage process, and weak planes were formed in the slope.

### 4.2. Relationship between the Apparent Resistivity and Water Content

Figure 7 shows the relationship between the apparent resistivity $\rho$ and water content $\theta$ in the rainfall-affected slope. The relationship between these parameters can be described by Equation (12) [37].

$$\rho = A_1 e^{(-\theta/t_1)} + A_2 e^{(-\theta/t_2)} + \rho_0 \tag{12}$$

where, $A_1$, $A_2$, $t_1$, $t_2$ and $\rho_0$ are constants.

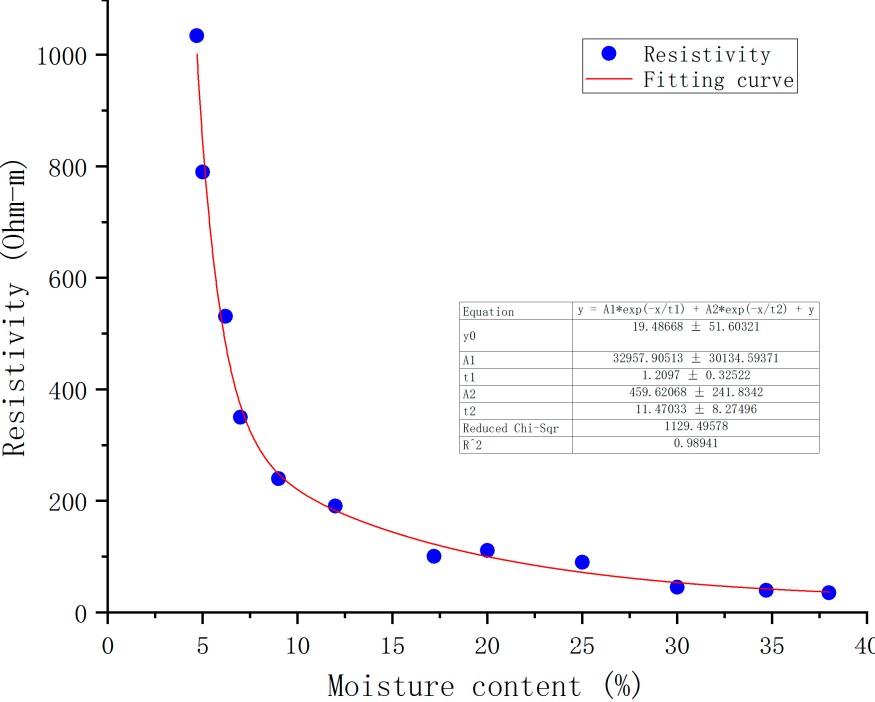

**Figure 7.** Relationship between apparent resistivity and water content.

The water content and saturation of the rainfall-affected slope increased with rainfall seepage. With increasing water content and saturation of the rock and soil mass, the charge mobility increased, and the apparent resistivity of the slope gradually decreased. Therefore, the change in the water content in the survey line area at the different stages can be described by the change in the apparent resistivity.

From Figure 7, it can be seen that the slope resistivity undergoes a process of a rapid decrease, a slow decrease, and then stabilization. At the beginning of rainfall, water in the slope transitioned from adsorbed water to capillary water with increasing water content, and the charge mobility increased. The conduction mode of the current changed from double electric layer conduction on the particle surface to double electric layer conduction and water-soil series joint conduction [38]. With increasing conductive path, the apparent resistivity of the slope rapidly decreased. As the water content continued to increase and the dump slope approaches saturation, the connectivity of the pore water in the rock and soil was enhanced. A current gradually came to be conducted by the pore water, and the rate of decrease in resistivity slows. After reaching the saturation state, the form of water changed to gravity water, and the current was mainly conducted through pore water. The influence of the increase in the water content on the resistivity was gradually weakened, and the resistivity tended to stabilize [39].

### 4.3. Development Trend of Disaster Sources in the Survey Line Area

According to the detection results of the high-density resistivity method, the resistivity change diagram of the rock and soil mass under the survey line was obtained by inversion, which describes the trends of water migration and crack formation in the slope. Typical time inversion results of the sections depicted in Figures 5 and 6 are shown in Figure 8, and the external changes in the slope at the corresponding time are shown in Figure 9.

Before the test, the resistivity of the slope was measured. The model material layer is uniform, the slope soil is relatively dry, and the slope showed a high resistance value. After the 5th rainfall event, the accumulated rainfall is 10.5 mm. The resistivity distribution in each layer within the slope survey line was obvious, and the resistance of the same horizontal layer was consistent. The depth of the wetting front is approximately 7 cm, and the resistance is approximately 65 $\Omega \cdot$m. The shallow slope contained an obvious low-resistance area, with the lowest resistivity of 19 $\Omega \cdot$m, as shown in Figure 8a. In Figures 5 and 6, the resistivity of the shallow slope experienced the first significant sudden change. After the 76th rainfall process, the accumulated rainfall was 159.6 mm, and preferential flow seepage was formed inside the slope, as shown in Figure 8b. After the 85th rainfall event, the accumulated rainfall was 178.5 mm, and several low-resistance areas were formed in the shallow slope of the survey line area, with the lowest resistance of 19 $\Omega \cdot$m. The abscissa ranged from 0.49 to 0.77 m, 1.07 to 1.29 m and 1.47 to 1.78 m, forming three obvious low-resistance areas. The low-resistance areas inside the slope moved downwards as a whole and showed uneven changes. The low-resistance area at the middle rapidly expanded, with a wetting front depth of 33.6 cm, as shown in Figure 8c. The resistivity of the deep slope showed a sudden change, as shown in Figure 6. After the 96th rainfall event, the accumulated rainfall was 201.6 mm, and the soil mass from the slope surface to a depth of 5 cm was saturated, with a minimum resistance of 19 $\Omega \cdot$m. The three low-resistance areas formed during the 85th rainfall event developed downwards. The low-resistance area from 1.07~1.29 m showed a low-resistance zone in the middle of the slope, indicating that serious water accumulation occurred in the slope and that the weak planes were well developed, as shown in Figure 8d. After the 100th rainfall process, the slope surface area at the abscissa from 1.14~1.19 m exhibited a relatively high resistance, with a resistance value of 61.5 $\Omega \cdot$m, and obvious cracks could be observed in electrode areas 16–18 along the corresponding slope surface measuring line. The sections from 0.63~1.05 m and 1.26~1.47 m showed relatively high resistance values, indicating that cracks were formed in corresponding areas inside the slope, as shown in Figure 8e. After the 111th rainfall event, the slope showed a low resistance as a whole. The resistance at

several locations within the depth range of 5.3~17 cm was higher than that of the soil at the same depth, indicating that cracks had formed in the slope, as shown in Figure 8f. The resistivity curves in Figures 5 and 6 show significant fluctuations.

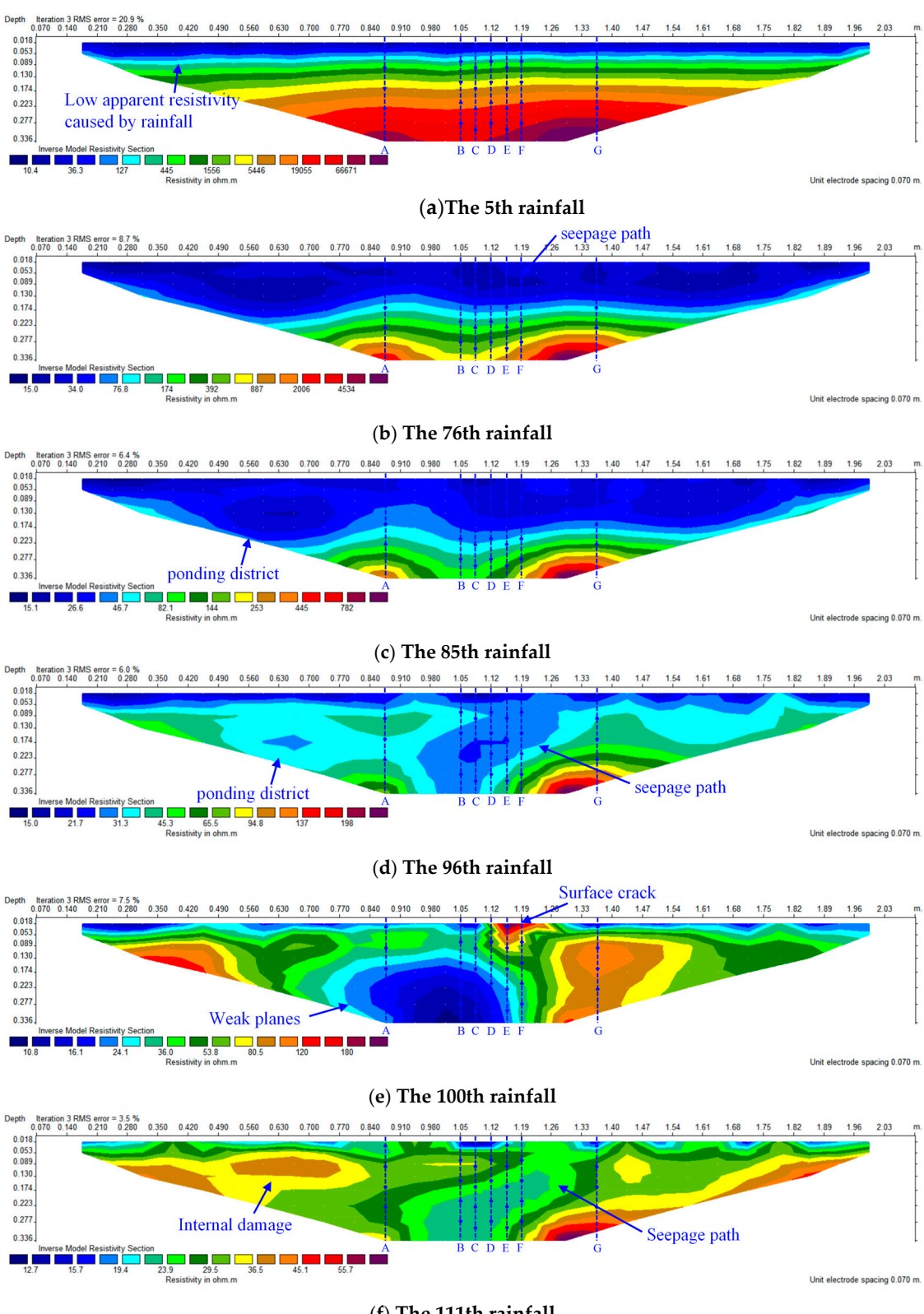

**Figure 8.** Inversion graph of apparent resistivity at typical time.

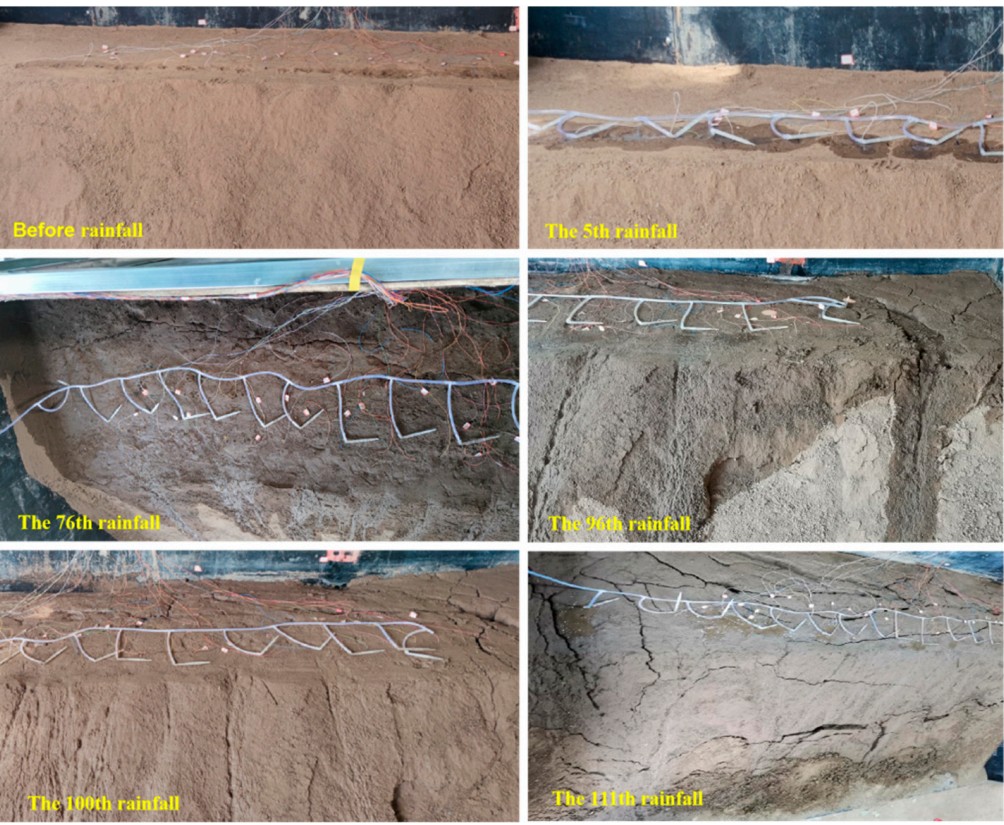

**Figure 9.** External changes to the rainfall slope at a typical time.

To observe the development process of cracks in the slope top, according to the change characteristics of the apparent resistivity and the development of fissures in the slope top, the resistivity from the 96th to 100th rainfall processes was selected for time-lapse resistivity inversion, as shown in Figure 10.

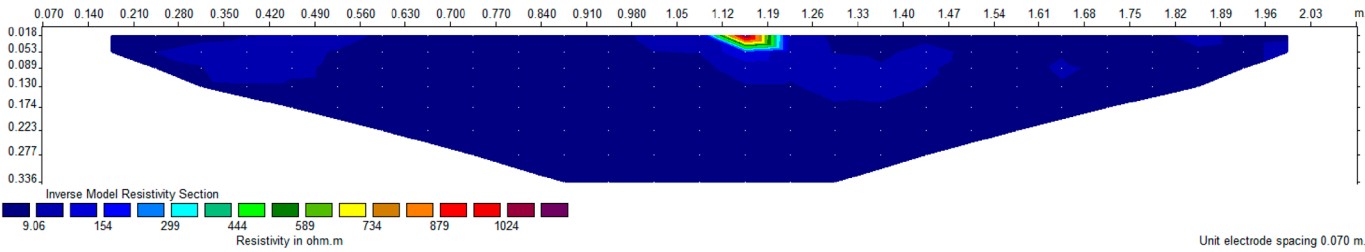

**Figure 10.** Time-lapse inversion diagram of internal damage in the slope from the 96th to 100th rainfall.

The residual value (Δρ) was selected as an index to describe the change in the resistivity during crack development. A positive Δρ value indicates the development of cracks, while a negative Δρ value indicates water accumulation inside the slope. The time-lapse resistivity inversion curve indicated that there was an obvious positive resistance abnormal change area from 1.05 m–1.19 m. The red area in the figure indicates that during the 96th to 100th rainfall cycle, the rock and soil mass in this area were damaged and cracks were formed. Excavation verification indicates that cracks had formed inside the slope in this area. From the inversion results, we could analyze the dynamic process from internal damage development of the slope to disaster source formation. This approach is more effective than conventional inversion and external change observations.

## 5. Discussion

### 5.1. Geoelectric Field Response Mechanism of Disaster Source Development

Wang et al., Jiang et al. and Li et al. [23–25], according to Maxwell's conductivity formula, derived equations of the resistivity change characteristics of a rock mass in different states under pressure, verifying the influence of rock porosity φ on resistivity $\rho_t$.

$$\rho_t = \rho_A \frac{\varphi}{3 - 2\varphi} \tag{13}$$

According to Equation (13), the resistivity of rock and soil is positively correlated with the porosity, which suggests that the larger the proportion of pores in rock and soil, the higher the resistivity. This conclusion is inconsistent with Archie's equation, which also shows that Archie's equation cannot suitably describe the resistivity changes in dry rock and soil masses. According to Equation (13), the development of pores (cracks) is the root cause of the change in the rock and soil mass resistivity, which can fully explain the process of the reduction in the apparent resistivity fluctuation of the rainfall-affected slope. After slope saturation, the relationship between the resistivity and porosity of the slope rock and soil mass can be readily obtained (Equation (14)). It is evident that the apparent resistivity of the saturated slope is negatively correlated with porosity, which is consistent with Archie's equation.

$$\rho_t = \rho_W \frac{3 - \varphi}{2\varphi} \tag{14}$$

The relationship between the apparent resistivity of the slope and the rainfall sequence (Figures 5 and 6) shows that the resistivity of the slope changes with changes in the water content and saturation of the slope. Archie's equation considers that the conductivity of a rock mass depends on the conductivity of the formation water contained in the pores, which can be used to describe the relationship between the resistivity and saturation of a pure rock mass without clay. However, Archie's equation ignores clay water and capillary water in the rock mass and treats the rock mass skeleton as an insulator, which deviates from the interpretation of the resistivity of an underground medium body [40].

The rock and soil mass of the rainfall-affected slope experienced a dynamic cycle process involving wetting and drying, in which the internal structure of the slope and the occurrence form of water changed, and the apparent resistivity significantly decreased. The three-water model [41,42] is based on Archie's equation [36,43] and the skeleton conductivity model [44]. The total conductive circuit of the rock and soil mass can be divided into a three-phase parallel connection of free water, microcapillary water and clay water based on the model (Figure 11). Zhang et al. and Fu et al. [45,46] introduced the lithology coefficient a based on the three-water model and achieved satisfactory results in the exploration and interpretation of low-porosity and low-permeability reservoirs in petroleum geology and sandstone and mudstone formations.

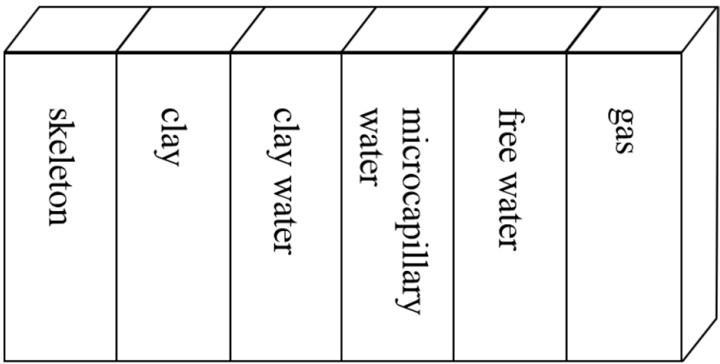

**Figure 11.** Water conduction model.

Free water is water that can freely flow along pores (cracks) under normal pressure, microcapillary water is water that exists in rock and soil micropores, and clay water is water with a specific conductivity formed under the action of cation exchange adsorption. The seepage characteristics of microcapillary water and clay water differ from those of free water, and these water types cannot flow and be produced under normal pressure. Table 5 shows the corresponding relationship between the three types of water in the model and the water in the rock and soil mass [47].

**Table 5.** Relationship of state between water in the water model and water in rock and soil masses.

| Three Water Model | Rock | Soil |
|---|---|---|
| Free water | Gravity water | Gravity water |
| Clay water | Bound water on clay surface | Bound water |
| Microcapillary water | Capillary water and bound water on skeleton surface | Capillary water |

When the rock and soil are unsaturated, the resistivity $\rho_t$ can be expressed as:

$$\rho_t = \frac{a_f \rho_{wf}}{\varphi_f^{m_f} S_{wf}^n} + \frac{a_i \rho_{wi}}{\varphi_i^{m_i}} + \frac{a_c \rho_{wc}}{\varphi_c^{m_c}} \tag{15}$$

$$S_w = \frac{\varphi_f S_{wf}^n + \varphi_i + \varphi_c}{\varphi} \tag{16}$$

where $\rho_{wf}$, $\rho_{wi}$ and $\rho_{wc}$ are the resistivities of free water, microcapillary water and clay water, respectively ($\rho_{wf} = \rho_{wi}$), $\rho_t$ is the resistivity of the unsaturated rock and soil mass, $a_f$, $a_I$ and $a_c$ are the lithology coefficients corresponding to the conductive network, $\varphi_f$, $\varphi_i$ and $\varphi_c$ denote the pore ratios of the three types of water and the total porosity of rock and soil, $m_f$, $m_i$ and $m_c$ are the cementation index values corresponding to the three types of water, $S_{wf}$ is the free water saturation in the pores in the unsaturated state, $n$ is the saturation index, and $S_w$ is the total saturation [46].

When the rock and soil mass are saturated, $S_w = 1$ and the resistivity $\rho_w$ can be expressed as:

$$\rho_w = \frac{a_f \rho_{wf}}{\varphi_f^{m_f}} + \frac{a_i \rho_{wi}}{\varphi_i^{m_i}} + \frac{a_c \rho_{wc}}{\varphi_c^{m_c}} \tag{17}$$

The three-water model indicates that the distribution state and conductive path of free water, microcapillary water and clay water differ, and the conductive mechanism also differs. They all follow Archie's equation. The model structure shows that the resistivity of water-bearing rock and soil is negatively correlated with the saturation and porosity. The conductivities of free water and microcapillary water are the same, which is related to the salinity of water. Clay water is less notably affected by salinity, and its conductivity is mainly due to the formation of hydration ions after the residual negative charge adsorbs cations in water, which is related to the temperature change. Both microcapillary water and clay water occur in the bound state, which is collectively referred to as bound water [46,47].

The three-water model was applied to describe the change in resistivity of the rainfall-affected slope. The behavior of rainfall-affected slopes can be regarded as a process of the mutual replacement of air and free water in slope pores (cracks). According to the three-water model, at the initial stage of rainfall seepage, the saturation of the rock and soil mass is low, and the aqueous solution enters the pores and is adsorbed onto the particle surface to form bound water, resulting in interface polarization. The dielectric constant rapidly increases with increasing saturation, and the slope resistivity abruptly decreases. With increasing saturation, free water increases and both the interface polarization phenomenon and the rock and soil mass conductivity further increase until water molecules enter the pores after exceeding the critical saturation, thus forming free water and subjecting the water molecules to polarization. Moreover, the increase in the dielectric constant decreases,

and the resistivity gradually stabilizes. Therefore, the apparent resistivity decline and mutation after the 5th and 76th rainfall events in this test could be explained as follows: the slope body was formed at the corresponding depth under the action of bound water in the rainfall seepage process; the apparent resistivity of the saturated rock and soil slowly decreased, which is mainly affected by free water.

The three-water model retains the free and bound water conductivity mechanism of the two-water model and considers the influence of the microcapillary water connectivity, which can effectively explain the sudden decline in the slope apparent resistivity in the rainfall process [40,41]. This test phenomenon is consistent with the research conclusions regarding the conductivity characteristics of low-resistivity sand shale oil and gas reservoirs [45].

Equations (13)–(17) explain the influences of pore (crack) development and saturation change on the resistivity, which can provide a petrophysical basis for resistivity method application and interpretation of the development process of geological disaster sources of rainfall-affected slopes.

### 5.2. Geoelectric Field Distribution Characteristics under Disaster Source Evolution

According to the inversion results of the resistivity before rainfall, there is a high-resistivity anomaly in the shallow area within the range of sections B–F. Figure 12 shows the evolution process of new cracks at the top of the slope and the high-resistance abnormal area. The comparison of the crack development in the slope top (Figure 12a) and the position of the high-resistance abnormal area (Figure 12b) reveals that new cracks are generated above and to the right of the high-resistance abnormal area. Due to the influences of the rock and soil composition, heterogeneity and other factors, the compactness of the high-resistance abnormal area is poor. Under the influence of seepage, small particles are easily lost, causing local damage. Stress unloading due to damage concentrates the stress in the adjacent low-resistance area, causing shear failure and forming new tension cracks. During seepage, shrinkage cracks inhibit water absorption and expansion, resulting in expansion cracks and an increase in resistivity. Shrinkage cracks are large in size and mostly formed at the early stage of the dry-wet cycle process, which are the main factors causing geological slope disasters. Expansion cracks are small in size and shallow in development, mostly formed at the late stage of the dry-wet cycle process, easily causing erosion damage in shallow areas [48].

According to Tang et al. [49], most cracks in soil are formed in the dry-wet cycle process after saturation. In this test, the resistivity in the high-resistivity area before rainfall fluctuated and decreased with increasing rainfall seepage process. After the 76th rainfall event, the resistivity of the soil at a depth of 0.175 m or more rapidly dropped due to saturation. After the 96th rainfall process, cracks could be detected in the slope interior, and after the 100th rainfall process, cracks could be observed in the slope top.

In summary, high-resistance abnormal areas and their adjacent areas of a slope before rainfall are potential geological disaster source-prone areas. In the damage process, the original high-resistance abnormal areas are gradually eroded under water infiltration, forming weak water-bearing areas with a low relative resistivity. Cracks are developed in the adjacent zones of these high-resistance abnormal areas, showing a relatively high resistance. The test conclusion is consistent with the previously reported research results [50].

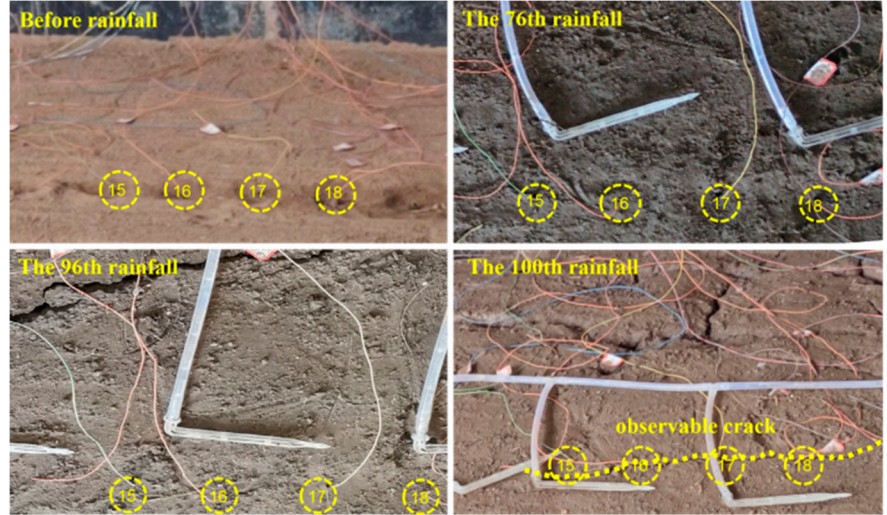

**(a)** cracks development on top of slope

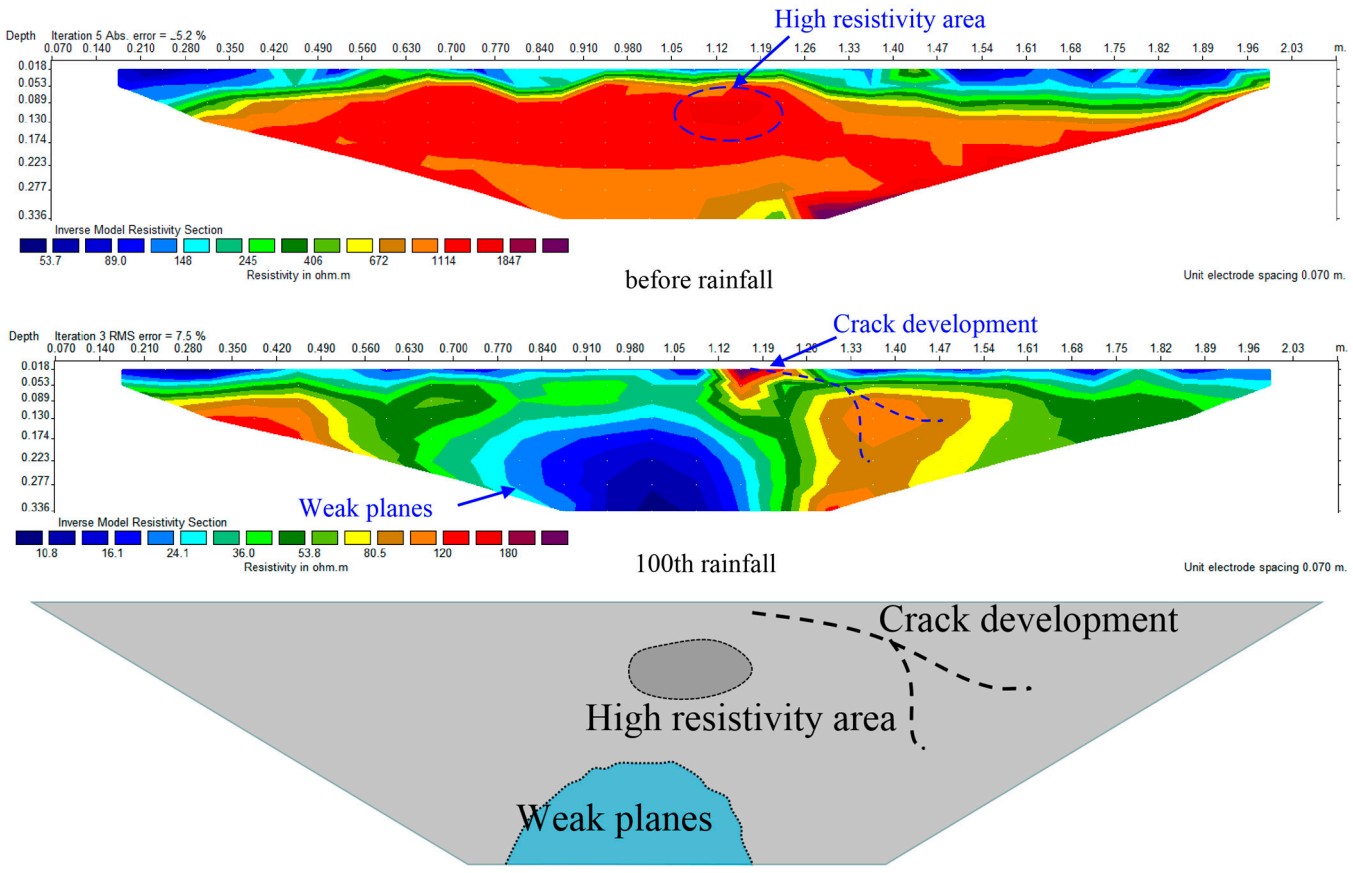

**(b)** crack development location

**Figure 12.** Comparison of development positions of disaster sources in the survey line area.

## 6. Conclusions

Based on the principle of the direct current resistivity method, a proportional model of a waste dump slope under rainfall seepage conditions was designed and produced. We studied the evolution and distribution characteristics of geological hazard sources in single-step waste dump slopes, derived a resistivity response mechanism equation for rainfall-induced slope hazards, and explored resistivity criteria for the formation of geological hazard sources in rainfall-affected slopes.

The results indicated that there is a significant negative correlation between the electrical resistivity and water content in the process of rainfall-induced slope damage. During the development of slope fractures, the resistivity increased, while during water infiltration, the resistivity decreased. Before obvious damage, such as observable cracks, emerged at the top of the slope, geological hazards, such as cracks and weak surfaces, had already formed within the slope, manifested as sudden changes and fluctuations in the apparent resistivity in the damaged areas. These sudden changes and abnormal fluctuations in the apparent resistivity could serve as a basis for identifying internal slope failure, and geological disasters could be predicted through relevant data features.

**Author Contributions:** E.K. wrote the main manuscript text, Z.Z. (Zexi Zhao), H.M. and Z.Z. (Zihao Zhao) are mainly responsible for the experimental part of the paper. All authors reviewed the manuscript. All authors have read and agreed to the published version of the manuscript.

**Funding:** University fund of Inner Mongolia Autonomous Region of China (NJZY19127), China's national science and technology innovation fund for College Students (202010127002). Inner Mongolia Natural Fund (2019MS04016).

**Institutional Review Board Statement:** Not applicable.

**Informed Consent Statement:** Not applicable.

**Data Availability Statement:** This article does not create new data.

**Conflicts of Interest:** The manuscript has not been published before and is not being considered for publication elsewhere. All authors have contributed to the creation of this manuscript for important intellectual content and read and approved the final manuscript. We declare there is no conflict of interest.

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
