# Peer review of "Distribution Characteristics of the Geoelectric Field in Waste Dump Slopes during the Evolution of Instability Sources under Rainfall Conditions"

_applsci, doi:10.3390/app13116459_

Round 1

Reviewer 1 Report

Dear Editor,

the overall quality of the manuscript is quite poor, the English form is sloppy and the methodological approach is simple and it is not in-line with the current state-of-the-art.  There is a great confusion on the use of 1D and 2D electrical resistivity methods and many other critical aspects.

Author Response

Dear experts and editors:

Thanks for every expert’s generous guidance. We have made careful revisions based on the suggestions, but there maybe still some unreasonable aspects. We will make further revisions based on your suggestions.

The structure and content of the manuscript have been optimized, and the manuscript has been re polished by native English language experts. The focus of this manuscript is to study the feature recognition and mechanism analysis of feasibility sources induced by rainfall changes. The methods and experimental processes used are consistent with those of other scholars in the references. We will choose more advanced technologies in future research to validate the experimental conclusions from different perspectives. The article uses the 2D electrical resistance method for image visualization analysis and feature analysis of resistance values in key areas. The full text does not use the 1D electrical resistance method.

More response please see the attachment 

Best regards,
Dr. Ensheng,Kang

Reviewer 2 Report

The manuscript presents a very interesting experiment on a physical model of soil slope under rainfall conditions. Scope of the research was the alocation of instability sources evolved in the soil mass due to the rainfall by measuring the electrical resistivity of the soil mass.

The topic is very important and the presentaion is scientifically sound. Authors could benefit from previous related publications from a number of case studies published by Sam Johansson, who applied electrical reisistivity method to investigate internal erosion, dam seepage, leakage detection etc. :  1) [https://www.researchgate.net/profile/Sam-Johansson-2]

2) [https://energiforsk.se/program/dammsakerhet/rapporter/seepage-monitoring-by-resistivity-and-streaming-potential-measurements-at-hallby-embankment-dam-2000-15/]

I am suggesting a change of the title as follow: 

"Distribution characterisitics of geoelectric field in waste dump slopes during the evolution of instability sources induced by rainfall"

Some corrections are necessary e.g.:

1) Correct "Aection" to "Section" in Figures 5, 6 of the above mentioned el.se

2) "[46] Archie, G. E. The electrical....J. Transactions of the American Institute of Mechanical Engineers, ..."

Figure 10 is very informative and could be compared with Figures 6 and 7 of "energiforsk.se" report mentioned above. 

Author Response

Dear experts and editors:

Thanks for every expert’s generous guidance. We have made careful revisions based on the suggestions, but there maybe still some unreasonable aspects. We will make further revisions based on your suggestions.

(2.1)Changed the title of the paper as "Distribution characteristics of the geoelectric field in waste dump slopes during the evolution of instability sources under rainfall"    line 2 - line 4

(2.2)Have corrected "Aection" to "Section" in Figures 5, 6   

(2.3)Have corrected References[46],new reference number is [43]  line 732

(2.4)A comparative analysis was conducted between Figures 5, 6 and Figures 10, and the corresponding positions in the text were modified.  Sections 4.3

More response please see the attachment 

Best regards,
Dr. Ensheng,Kang

Reviewer 3 Report

#Review

Article: Distribution characteristics of geoelectric field during the evolution of geological disaster sources on rainfall slope

General appreciation

In general terms the article titled “Distribution characteristics of geoelectric field during the evolution of geological disaster sources on rainfall slope”, reveals interest and usefulness. The physical model of rainfall slope was designed and made and it is well explained and presented, as well as the methodology. The Introduction, although well founded, should include some more current references related to. The results are discussed in detail and clearly. There are still some aspects that could desearve some attention and which will be mentioned below.

ü  As it is a large article at the end of the Introduction chapter, perhaps it would be convenient to briefly and succinctly present the chapters that comprise it.

ü  In Figure 1, it would be appropriate to reorganize its own legend relative to points 1 to 8. If possible, it is suggested to increase the size of the two images and the schematic representation.

ü  Tables 3 and 4 are practically glued together, there should be some text separating them.

ü  The title of Chapter 3 could be different, something more generic and not so concrete, such as “theoretical concepts”.

ü  The different images in Figure 2, 3, 10, 11, 13, 15 should be centered.

ü  In the Bibliography, references [18] and [34] are undated.

 [18]Li, X.W.; Zhou, H.X; Che, A.L. Application of resistivity method in the study of damage accumulation of rock slope with weak intercalation. 805 J/OL. Journal of Engineering Geology:1-10.

 [34]Xie, T.; Lu, J. Apparent resistivity anisotropic variations in cracked medium. Chinese J. Geophys. 63(4):1675-1694.

Author Response

Dear experts and editors:

Thanks for every expert’s generous guidance. We have made careful revisions based on the suggestions, but there maybe still some unreasonable aspects. We will make further revisions based on your suggestions.

(3.1) Have added recent references in the introduction.  references:[1],[9],[10],[11]

(3.2) At the end of the introduction, a chapters introduction to the main content of the article has been added   line 96 - line 100

(3.3) Added the legend to Figure 1 and modified the dimensions and scale in  Figure1.   

(3.4) The location and description of Table 3 and Table 4 are adjusted. 

(3.5) Revise the title of Chapter 3 to “Fundamentals”  line 196

(3.6) The images in Figure 2, 3, 10, 11, 13, 15 have been centered

(3.7) Supplemented the dates of references [18] and [34]  

More response please see the attachment 

Best regards,
Dr. Ensheng Kang

Reviewer 4 Report

The authors have presented an interesting article. However, it is quite long and needs to be shortened:

1. Section 4.1 and 4.2 is too long. Maybe a good solution is the presentation of geoelectric field characteristics of disaster sources evolution using a diagram. The experiment results can be presented in a table with a short description.

2. Are all presented formulas and theoretical descriptions necessary to explain this experiment?

3. The conclusions should contain only the most important results without duplicating previous content.

And the last question, Did the authors consider temperature and time when doing the experiment? Do external factors such as weathering affect the formation of cracks?

Author Response

Dear experts and editors:

Thanks for every expert’s generous guidance. We have made careful revisions based on the suggestions, but there maybe still some unreasonable aspects. We will make further revisions based on your suggestions.

(4.1)The manuscript has been shortened.  Section 4.1was shortened,  Section 5.2 was deleted.

(4.2) We tried to express the test results in chart, but it is not convenient to explain them accordingly when describing the results. Therefore, we reseted 4.1 and deleted “ the resistivity change with depth” .

(4.3) Equation 1 describe research background of rainfall dump slopes, and it is recommended to retain it for the completeness of the research. The other Equations have substantial applications in explaining experimental phenomena and calculations.

(4.4) The conclusion has been simplified as required.

(4.5) Simplified assumptions were made in this experiment, without considering the effects of temperature, time, and weathering. Only the variation of resistivity with rainfall was studied. In future research, we will delve into the changes of waste dump under the coupling influence of other factors.

More response please see the attachment 

Best regards,
Dr. Ensheng Kang

Round 2

Reviewer 1 Report

Dear authors,

I have appreciated your reply  and your effort for improving the quality of manuscript. However, the quality of the overall organization is always questionable. I strongly suggest to introduce more details in figure captions and to better describe the results displayed on the figures. Furthermore, an accurate revision of the English form is mandatory.

Author Response

(1)Revised Figure captions in Figures 5, 6, 10, 11, and 12 to include detailed descriptions.

(2)Supplementary explanations have been provided to Figures 9 and 12 to illustrate the variation of electrical resistivity with water content and the process of slope damage

(3)Revised the details and English of the form

(4)We have checked English language, style and spell. Invite native English experts to revise the language and writing details of the entire article

Reviewer 4 Report

All previous comments from the reviewer have been taken into account by the authors.

Author Response

(1)Revised Figure captions in Figures 5, 6, 10, 11, and 12 to include detailed descriptions.

(2)Supplementary explanations have been provided to Figures 9 and 12 to illustrate the variation of electrical resistivity with water content and the process of slope damage.

(3)Revised the details and English of the form.

(4)We have checked English language, style and spell. Invite native English experts to revise the language and writing details of the entire article.

Round 3

Reviewer 1 Report

Dear authors,

the overall quality of the manuscript was improved, but some fundamental problems still remain.

There is a great confusion in the use of the term "apparent resistivity".

The apparent resistivity values are the results of the experimental observations, while the resistivity values are the results of the 2D data inversion.

In the paper, the term "apparent resistivity" and the term "resistivity" are often used incorrectly.  If the values reported in the figures 5 and 6 are "apparent resistivity values", it is not necessary introduce the inversion theory. On the contrary, if the values reported in the figures are the result of the 2D inversion they represent "resistivity values".

This difference is substantial in any paper concerning the geoelectrical method!

I  strongly suggest to remove this ambiguity in the  manuscript.

Author Response

Thank you for your valuable comments. We have made modifications to the ambiguity in the application of resistivity and apparent resistivity in this paper based on your suggestion. Your suggestions is very helpful to this research, and your rigorous attitude is worth to learn for us. We would like to express our gratitude for your guidance once again.

Please refer to the revision position in the paper for specific modifications.
